:◯: PLOS | ONE

# Affective state determination in a mouse model of colitis-associated colorectal cancer

Lauren C. Chartier [1,2], Michelle L. Hebart[3], Gordon S. Howarth[2,3], Alexandra L. Whittaker [3◉], Suzanne Mashtoub[1,2,4◉]*

**1** Discipline of Physiology, Adelaide Medical School, The University of Adelaide, Adelaide, South Australia, Australia, **2** Department of Gastroenterology, Women's and Children's Hospital, North Adelaide, South Australia, Australia, **3** School of Animal and Veterinary Sciences, The University of Adelaide, Roseworthy Campus, Roseworthy, South Australia, Australia, **4** School of Medicine, The University of Western Australia, Murdoch, Western Australia, Australia

◉ These authors contributed equally to this work.
* suzanne.mashtoub@adelaide.edu.au

**Data Availability Statement:** All relevant data are within the manuscript and its Supporting Information files.

## Abstract

Behavioural indicators of affective state, including burrowing, clinical scores and the Mouse Grimace Score have not yet been validated in mouse models of chronic gastrointestinal disease. Additionally, a comparison of these methods has not been characterised. This study aimed to determine which behavioural assessment was the optimal indicator of disease, evidenced by correlation with clinically-assessed measures, in an azoxymethane (AOM)/dextran sulphate sodium (DSS) mouse model of colitis-associated colorectal cancer. C57BL/6 mice were allocated to four groups (n = 10/group); 1) saline control, 2) saline+buprenorphine, 3) AOM+DSS+water, 4) AOM+DSS+buprenorphine. Mice were gavaged thrice weekly with water or buprenorphine (0.5mg/kg; 80µL) for 9 weeks. Disease activity index (DAI) was measured daily; burrowing and grimace analyses occurred on days -1, 5, 19, 26, 40, 47 and 61. Colonoscopies were performed on days 20, 41 and 62. All animals were euthanized on day 63. Burrowing activity and retrospective grimace analyses were unaffected ($P>0.05$), whilst DAI was significantly increased ($P<0.05$) in mice with colitis-associated colorectal cancer compared to normal controls. In addition, DAI was positively correlated with colonoscopically-assessed severity and tumour number ($P<0.05$). We conclude that traditional measures of DAI or clinical scoring provide the most reliable assessment of wellbeing in mice with colitis-associated colorectal cancer.

## Introduction

Pain, as defined by the Oxford dictionary, refers to a 'highly unpleasant physical sensation caused by illness or injury'. In biomedical research, rodents are the most widely used species and it is estimated that globally approximately 4.6 million will experience procedure-related pain [1]. Prevention and alleviation of pain through accurate pain assessment and appropriate analgesic use should be a priority for researchers working with laboratory animals [2]. However, assessment of pain is challenging in all animal species, and is particularly problematic in rodent-prey species that mask pain as part of a survival mechanism [3].

**Funding:** LCC received partial funding from The Australian Veterinary Association for the current study (Animal Welfare Trust Grant; www.ava.com.au). LCC is also supported by an Australian Government Research Training Program Stipend and a PhD Top-Up Scholarship from AgriFutures Australia (www.agrifutures.com.au). The funders had no role in study design, data collection and analysis, decision to publish, or preparation of the manuscript. There was no additional external funding received for this study.

**Competing interests:** The authors have declared that no competing interests exist.

Although directly measuring pain in animals is near impossible, it can be presumed that animals are in pain when they display pain-like behaviours [4]. Such behaviours include reduced ambulation, agitation and increased grooming of an affected area [4]. A number of techniques have been established to measure pain-like behaviour in animal models. The first included stimulus-evoked measures such as the Von Frey, Randall-Selitto and Hargreaves techniques. These methods are now used less widely due to a growing concern over clinical translatability [4], since these methods are regarded as not measuring the affective pain response. In response to this concern, scientists developed a range of behavioural assessment methods proposed to measure the affective or emotional component of the pain response. A method that has received much attention is the characterisation of facial expression.

The first standardised system for facial expression scoring in rodents, 'The Mouse Grimace Scale' (MGS) was developed by Langford et al. (2010). The MGS scores five facial features or 'action units' from 0–2 (not present to severe). These features are: orbital tightening, nose bulge, cheek bulge, ear position and whisker change. A higher MGS score is indicative of increased levels of pain [5]. Whilst this system represents a considerable advancement in pain assessment of rodents, validation studies have typically involved retrospective assessment through analysis of stored video behavioural data in models of acute pain. Consequently, refinement possibilities are limited, since humane endpoints and analgesic provision are not able to be immediately implemented. Therefore, an alternative live-scoring method should be considered to allow 'cage-side' analysis, whereby interventions can be applied by researchers to rapidly improve animal welfare as needed. Leung et al. (2016) determined that a real-time grimace scoring method was reliable in rats [6]. Miller and Leach (2016) investigated the validity of baseline grimace scoring in various cohorts, strains and sexes of mice [7]; however, the effectiveness of real-time scoring in mice with chronic disease is yet to be determined. Furthermore, there have been relatively few investigations into the validity of MGS in mice expected to be experiencing *chronic* visceral pain, as opposed to acute pain, initiated via a non-surgical insult.

In addition to pain, animals may also experience distress or sickness leading to a negative affective state and potentially compromising their welfare. Negative affective state has traditionally been assessed in biomedical research using general clinical scoring, for example the Morton and Griffiths (1985) schema. This scheme describes appearance, food/water intake, behaviour, digestive and cardiovascular signs on a scale of normal to severe for rodents, guinea pigs, rabbits, cats and dogs [8]. This method remains the predominant method for laboratory rodent welfare assessment globally, as prescribed by animal ethics committees and regulatory documents. More recently, deterioration in activities of daily living (ADL) has been proposed to indicate decreased welfare in mice [9]. The most common measurable ADL in mice are burrowing, nesting and hoarding. These techniques are inexpensive, simple to run and also provide environmental enrichment for laboratory mice.

The current study sought to address these deficiencies in knowledge by determining the effectiveness of a range of measures of pain and well-being in a pre-clinical setting of colitis-associated colorectal cancer, using the azoxymethane (AOM)/dextran sulphate sodium (DSS) mouse model. Methods examined were the MGS, burrowing activity and clinical scoring and we aimed to determine which method was the most reliable in this pre-clinical model of colitis-associated colorectal cancer. Buprenorphine, a long-lasting opioid analgesic (up to 8 hours), has few effects on the immune system and has displayed efficacy in reducing the acute and chronic pain experience of mice and rats [10–12]. Therefore, buprenorphine was administered to validate the tests, especially the pain-specific MGS, to determine if pain was a contributing factor in behavioural response. The current study represents the first validation of a live-scoring method of the MGS, compared to the traditional retrospective scoring, in a mouse

model of chronic disease. Finally, this study aimed to determine the most reliable behavioural assessment technique (MGS, clinical scoring or burrowing) for indication of disease and its progression in experimentally-induced colitis-associated colorectal cancer, as evidenced by correlation with clinically-assessed disease measures in mice.

## Materials and methods

### Animal studies

All animal studies were conducted in compliance with the Australian Code for the Care and Use of Animals for Scientific Purposes and were approved by the Animal Ethics Committee of the Children, Youth and Women's Health Service (AE1095/7/21). This study was conducted as part of another study evaluating naturally-sourced therapies in colitis-associated colorectal cancer with control groups being utilised in the current study (AE1079/3/21). Female C57BL/6 mice (C57BL/6JArc, n = 40; average weight 18.36g) at 8 weeks of age were sourced from a SPF production facility, the Animal Resource Centre (ARC; Perth, Western Australia) and group-housed in standard open-top cages (polypropylene; 470mm x 175mm x 120mm; Crestware Industries) with pelleted paper bedding materials (>99% recycled paper product; Fibrecycle PtyLtd, Helensvale, QLD, Australia). The ARC undertakes a quarterly health screening, covering various bacterial, viral and parasitic organisms, all of which the obtained colony screened negative for. Only female mice were used to remain consistent with data obtained from previous studies [13, 14]. Environment was regulated at 21-24°C with 42–44% humidity and a light/dark cycle of 14:10 h. Mice were fed standard mouse chow (meat free mouse diet; Specialty Feeds, Glen Forrest, Western Australia) and provided with enrichment items including shredded paper, polycarbonate 'houses' and cardboard toilet paper rolls for the duration of the trial. All mice received *ad libitum* access to plain drinking water during the experimental period (except where group allocation precluded it).

### Experimental design

Female C57BL/6 mice (n = 10/group) were randomly assigned to four treatment groups (n = 10/group); 1) saline + water + water, 2) saline + water + buprenorphine, 3) AOM + DSS + water and 4) AOM + DSS + buprenorphine. Mice were stratified to groups based on baseline body weight. Group size (n = 10/group) was calculated using Clin.Calc for mouse grimace scale outcomes from Rosen et al. (2017) [15]. This calculation assumed a power of 80%, and indicated that a minimum sample size of n = 9/group was necessary for statistical power. All mice were administered (oral-gastric gavage) 80μL of either water or buprenorphine (0.5mg/kg; Reckitt Benckiser Healthcare, Hull, U.K) thrice weekly for the duration of the trial. Buprenorphine was administered via oral gavage as control animals (groups 1 and 3) utilised in another study were gavaged with water and thus exposed to the same procedural distress. On day 0, mice received a single intraperitoneal injection of saline or AOM (7.4mg/kg; average injection volume 0.14ml; 27G needle;Sigma-Aldrich, Castle Hill NSW, Australia), and then underwent three DSS/water cycles comprised of 7 days DSS (*ad libitum*; 2%w/v, 2g/100ml distilled water; MP Biomedicals LLC, Santa Ana California, USA) followed by 14 days plain water *(ad libitum)*. Negative control animals (groups 1 & 2) received plain water in their drinking bottles for the duration of the 9-week experimental period. All animals were euthanised on day 63 via $CO_2$ asphyxiation followed by cervical dislocation (experimental timeline; Fig 1).

### Disease activity index

DAI was calculated daily (at 8am, prior to buprenorphine administration) from general clinical signs including bodyweight loss, general condition, stool consistency and rectal bleeding

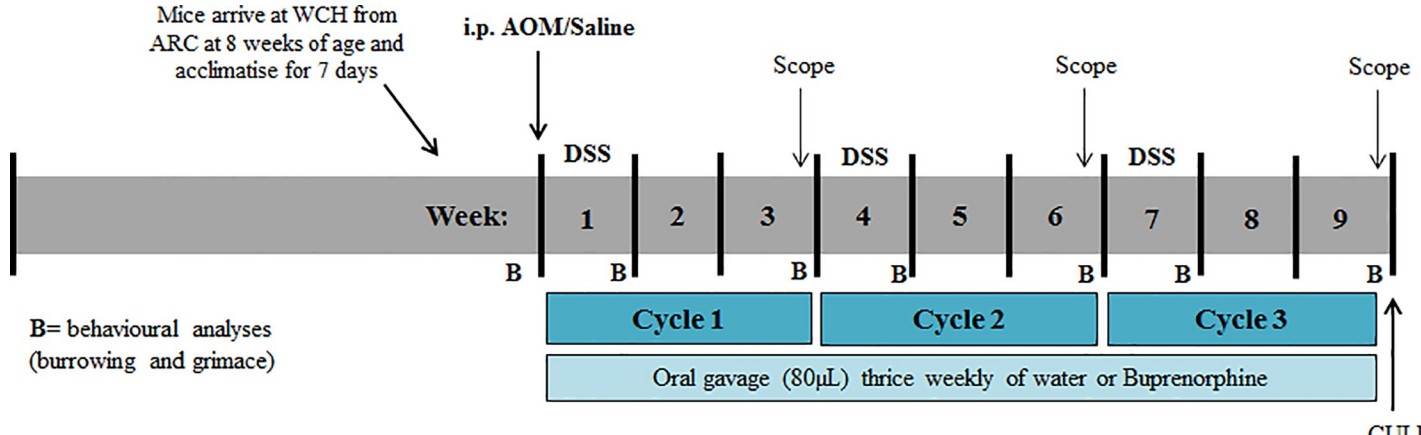

**Fig 1. Experimental timeline.** Animals were injected (i.p.) on day 0 and underwent three dextran sulphate sodium (DSS)/water cycles, comprising one week 2% DSS followed by 2 weeks of plain water. Animals were gavaged thrice weekly with water or buprenorphine. All mice were euthanised after 9-weeks via $CO_2$ asphyxiation, followed by cervical dislocation.

during the experimental period. General condition included features such as ruffled coat and grooming, hunching, alertness and abdominal twitching. Each parameter was scored from 0–3 with increasing severity and totalled to give a final DAI value, with a maximum possible score being 12 [16]. As DAI was a part of daily monitoring and welfare measurements, the researchers were not blinded to treatment groups when obtaining DAI scores.

## Colonoscopy

Colonoscopies using a high-resolution Karl Storz colonoscope (1.9mm outer diameter, Tuttlingen, Germany) were performed at the end of each DSS/water cycle (days 20, 41 and 62) to assess colitis progression and tumour development. Mice were anaesthetised using isoflurane inhalant (AbbVie Inc, Illinois, USA) in oxygen via mask for the duration of the procedure, and closely monitored on a heating pad during and immediately following the procedure. From anaesthetic induction to recovery, the colonoscopy procedure lasted approximately 10 minutes. Colitis severity was measured from recorded videos in a blinded fashion using five parameters described by Becker et al. (2005). These parameters include; thickening of the colon, vasculature pattern, presence of fibrin, granularity of mucosal surface, and stool consistency. Each parameter was scored from 0 to 3 with increasing severity and totalled, with the maximum possible severity score being 15 [17]. Additionally, colonic tumours were also counted from videos in a blinded fashion.

## Burrowing analyses

Burrowing analyses were conducted as a measure of affective state or activities of daily living using a modified protocol described by Deacon [18]. At 6pm, one hour after commencement of the dark cycle and 8 hours after buprenorphine administration (from 6pm), mice were placed in individual cages with a pre-weighed (400g kitty litter 'pebbles'; Black and Gold, Australian Asia/Pacific Wholesalers Pty Ltd) burrow attached (modified plastic Coca-Cola bottle; 6.9cm diameter, 16cm long) and left for an hour. After this time, the burrows were re-weighed and the weight difference taken to represent the amount burrowed. Burrowing analyses occurred on day -1 (baseline), at the end of each DSS week for a severe disease measure (days 5, 26 and 47) and at the end of each DSS/water cycle to assess recovery (days 19, 40 and 61).

## Mouse grimace scale

The affective experience of pain was assessed using the Mouse Grimace Scale (MGS; [5] at baseline, the end of each DSS week and end of a DSS/water cycle (days -1, 5, 19, 26, 40, 47 and 61). Real-time [6] and retrospective [5] MGS scoring methods were conducted in the morning (following buprenorphine administration; approx. 9am-12pm) at all indicated time-points. Five facial features (orbital tightening, nose bulge, cheek bulge, ear position and whisker change) were scored by a treatment-blinded grimace experienced observer from 0–2 (not present to severe), with a maximum possible MGS score being 10.

## Real-time MGS

Animals were removed from their home cage and placed individually in a clear plastic cage for scoring by a treatment-blinded experienced observer. The mouse remained in the scoring cage for a five minute period, where the observer assigned a score for each facial feature every 15 seconds. The animal was then returned to its home cage. A median score was calculated for each parameter per 15 second time-point and then an average was obtained of the medians per 90 second period. A final mean was then calculated from each 90 second period to produce a final grimace score for each mouse.

## Retrospective MGS

Over the same time period as the real-time method, video recording of the clear cage was performed using two video cameras placed on perpendicular cage sides (Panasonic HC-V180, Osaka, Japan). Still images of the mice were extracted from video footage, cropped to show the face alone and placed into a pre-designed excel spreadsheet by an investigator blinded to treatment allocation. A random number generator was used to select three images for scoring of each mouse at each time-point. These images were then scored by a treatment-blinded scorer using the methods described by Langford et al. 2010. Scores for each parameter were totalled to give a score per photo, and then the three photo scores were averaged to give a final reportable score for each mouse per time-point [5].

## Statistical analysis

Statistical analysis was completed using SPSS, version 25 for Windows (SPSS Inc. Chicago, Illinois, USA). Data were tested for normality using a Shapiro–Wilk test. DAI, burrowing activity, colitis score, and tumour number were analysed by repeated measures ANOVA with least significance difference (LSD) to compare among and within a group. MGS data were analysed non-parametrically using a Friedman test to assess temporal differences within groups and a Kruskal-Wallis with a Mann-Whitney *post-hoc* test to compare differences between groups within time-points. To determine any correlations between behavioural outputs and the measured clinical parameters, a non-parametric spearman-rho test was applied. $P < 0.05$ was considered statistically significant.

# Results

## Disease activity index

In normal animals, buprenorphine administration did not impact DAI scores during the experimental period compared to saline controls ($P > 0.05$; Fig 2). AOM/DSS significantly increased DAI scores on days 2, 3 and 5–63 compared to untreated saline controls ($P < 0.05$). In AOM/DSS mice, buprenorphine administration increased DAI scores on days 16, 18–21, 25, 41 and 59 compared to AOM/DSS controls ($P < 0.05$). Furthermore, buprenorphine

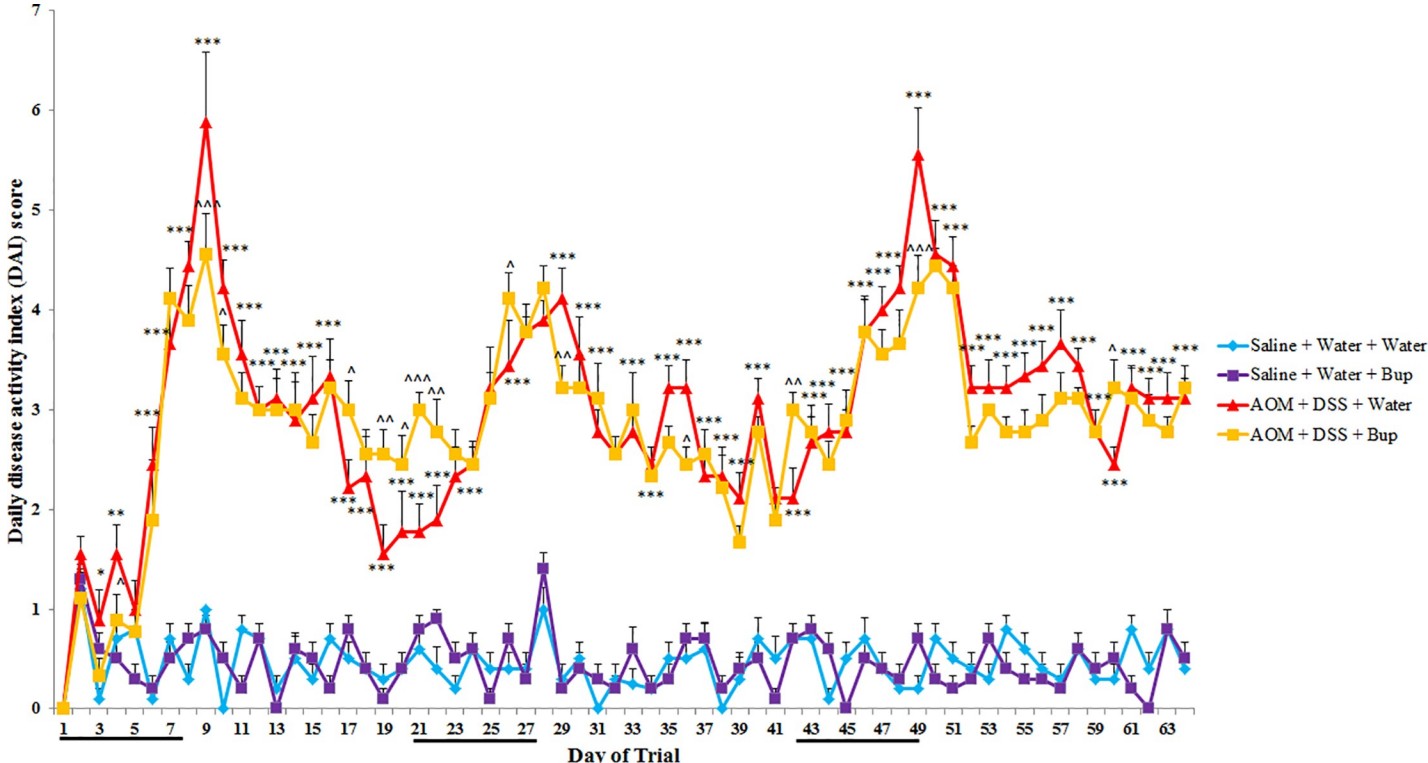

**Fig 2. Daily disease activity index (DAI) score (n = 10/group).** Data were analysed using a repeated measures ANOVA with least significance difference (LSD) and are expressed as mean DAI score ± SEM (black line on the x axis represents a dextran sulphate sodium; DSS week). ***$p<0.001$, **$p<0.01$, *$p<0.05$ compared to Saline + Water + Water, ^^^$p<0.001$, ^^$p<0.01$, ^$p<0.05$ compared to AOM + DSS + Water.

administration decreased DAI scores on days 3, 8, 9, 28, 35 and 48 in AOM/DSS animals compared to AOM/DSS alone ($P<0.05$).

## Colitis severity and tumour number

Buprenorphine administration did not impact colitis progression in saline control animals throughout the experimental trial ($P>0.05$; Fig 3). AOM/DSS significantly increased colonoscopically-assessed colitis severity compared to saline controls at all three time-points (days 20, 41 and 62; $P<0.05$). Mice administered buprenorphine and treated with AOM/DSS presented with increased colitis severity on day 20 and decreased colitis severity scores on day 62 compared to AOM/DSS controls ($P<0.05$).

Saline control animals and those treated with buprenorphine did not develop colorectal tumours during the experimental period. AOM/DSS resulted in increased tumour number compared to saline controls ($P<0.05$; Fig 4). Additionally, in AOM/DSS mice, buprenorphine did not significantly impact tumour development compared with AOM/DSS controls ($P>0.05$).

## Burrowing

Buprenorphine administration significantly increased burrowing activity in normal mice compared to saline controls on days 19, 26 and 40 ($P<0.05$; Fig 5). AOM/DSS did not significantly affect burrowing compared to saline controls at any time-point.

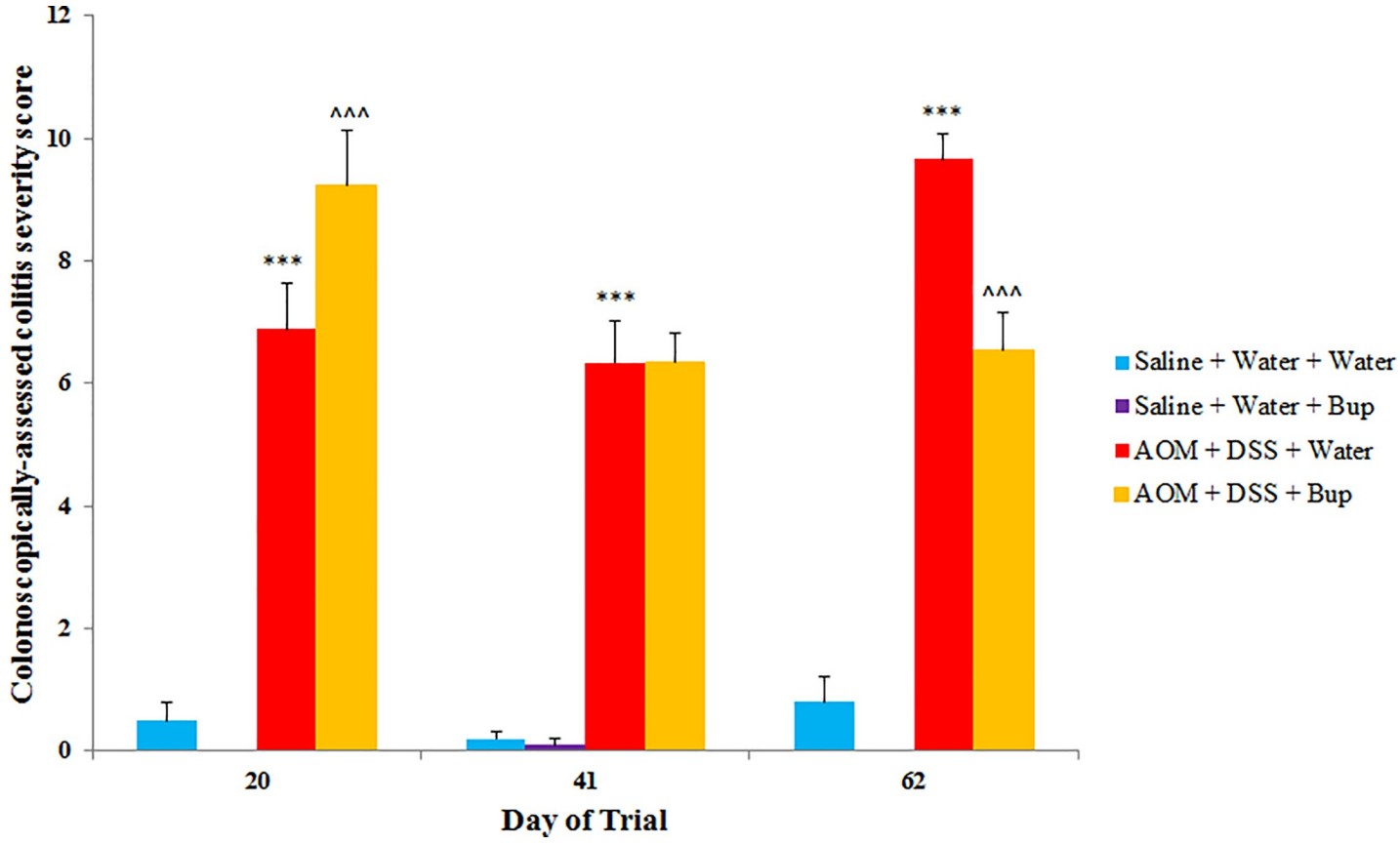

**Fig 3. Colonoscopically-assessed colitis severity (n = 10/group).** Data were analysed using a repeated measures ANOVA with least significance difference (LSD) and are expressed as mean colitis severity score ± SEM. Colitis-severity is calculated from stool consistency, mucosal thickness, granularity of the mucosal surface, fibrin and vasculature pattern (each scored from 0–3 and summed). ***$p<0.001$, **$p<0.01$, *$p<0.05$ compared to Saline + Water + Water, ^^^$p<0.001$, ^^$p<0.01$, ^$p<0.05$ compared to AOM + DSS + Water.

### Mouse grimace scale

Buprenorphine administration in normal mice resulted in no significant differences in real-time grimace scores at any time-point when compared to saline controls (Table 1; $P>0.05$). On day 19, AOM/DSS controls had higher real-time grimace scores compared to saline controls (Table 1; $P<0.05$), with no other significant differences on other days. AOM/DSS controls presented with significantly higher real-time grimace scores on day 19 compared to baseline ($P<0.05$). Buprenorphine administration in AOM/DSS mice resulted in significantly higher real-time grimace scores on day 19 when compared to baseline; and on day 47 compared to day 40 ($P<0.05$). Finally, buprenorphine administration in AOM//DSS mice significantly reduced real-time grimace scores on day 40 compared to day 19 ($P<0.05$). Scoring of retrospective grimace frames resulted in no significant differences within or across groups ($P>0.05$).

### Correlations

Real-time grimace scores were positively correlated with colitis severity and tumour number on day 19 (Table 2; $P<0.05$). Burrowing was negatively correlated with colitis severity and tumour number at all three time-points (days 19, 40 and 61; $P<0.05$). Furthermore, DAI was

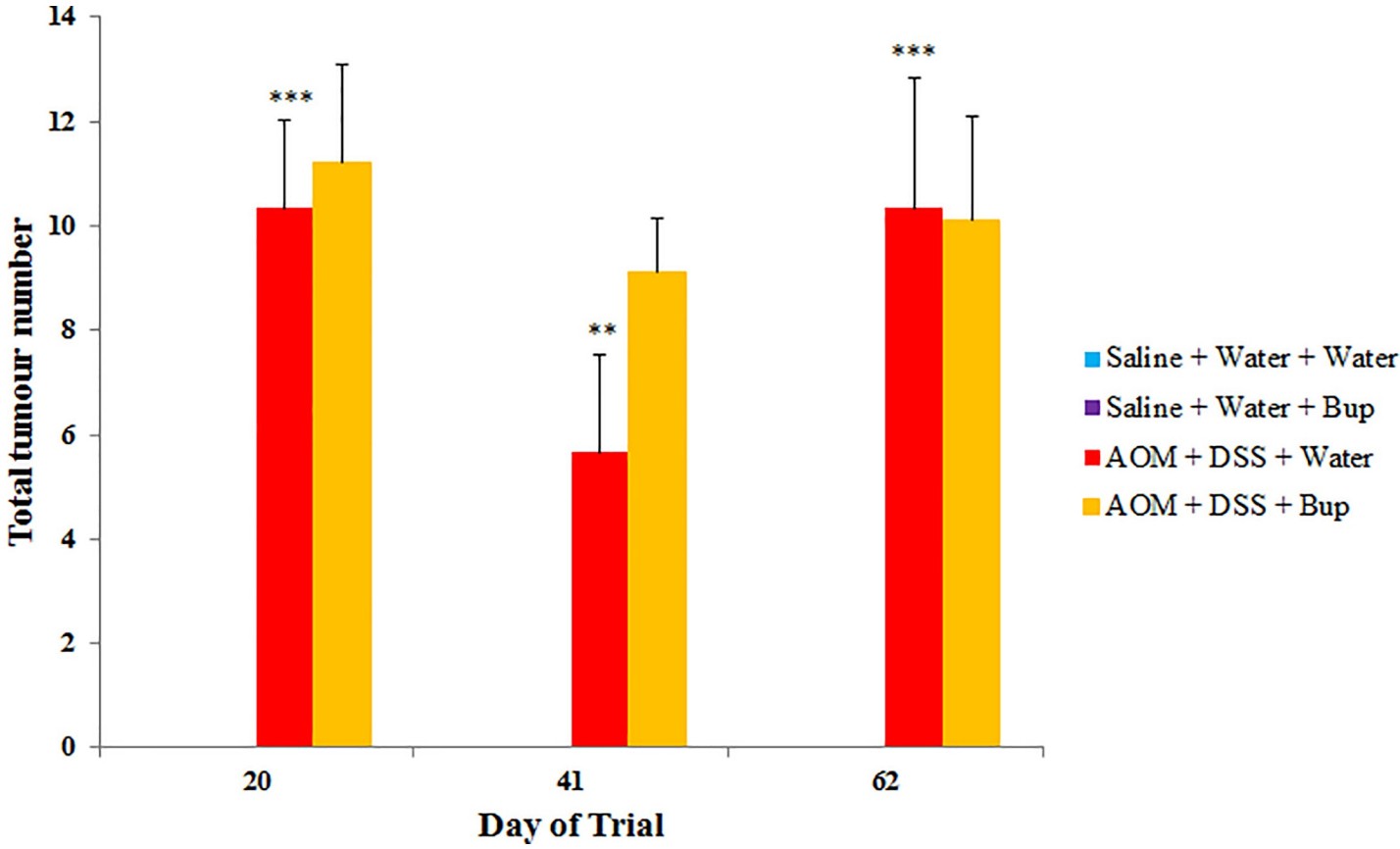

**Fig 4. Total tumour number measured from colonoscopy (n = 10/group).** Data were analysed using a repeated measures ANOVA with least significance difference (LSD) and are expressed as mean total tumour number ± SEM. ***$p < 0.001$, **$p < 0.01$, *$p < 0.05$ compared to Saline + Water + Water, ^^^$p < 0.001$, ^^$p < 0.01$, ^$p < 0.05$ compared to AOM + DSS + Water.

positively correlated with colitis severity score and tumour number at all three time-points (days 19, 40 and 61; $P < 0.05$).

## Discussion

AOM/DSS administration successfully induced colitis-associated colorectal cancer in mice, as evidenced by colonoscopically-assessed severity, tumour development and increased colon weights. However, the disease state was not reliably translated in the results of the two affective state measurement techniques utilised, namely burrowing and MGS. Clinical scores of disease such as DAI used in the current study, include scoring of non-specific mouse illness signs such as bodyweight loss, coat appearance, activity and stool consistency. Our findings suggest that the DAI is in fact the most reliable determinant of the clinical picture presented in these mice, and humane endpoint implementation in this model should continue to be based on this scoring scheme.

Analgesic administration did not affect normal animals; however, interestingly, buprenorphine increased the clinical DAI score of mice with colitis-associated colorectal cancer on selected days, attributed to bodyweight loss. This result was likely to be primarily due to in-appetence, possibly brought about by nausea, and consequent bodyweight loss as a side-effect of analgesic intervention [1, 19]. Nonetheless, this effect was not observed consistently throughout the experimental period. Overall, results were unable to conclude a significant

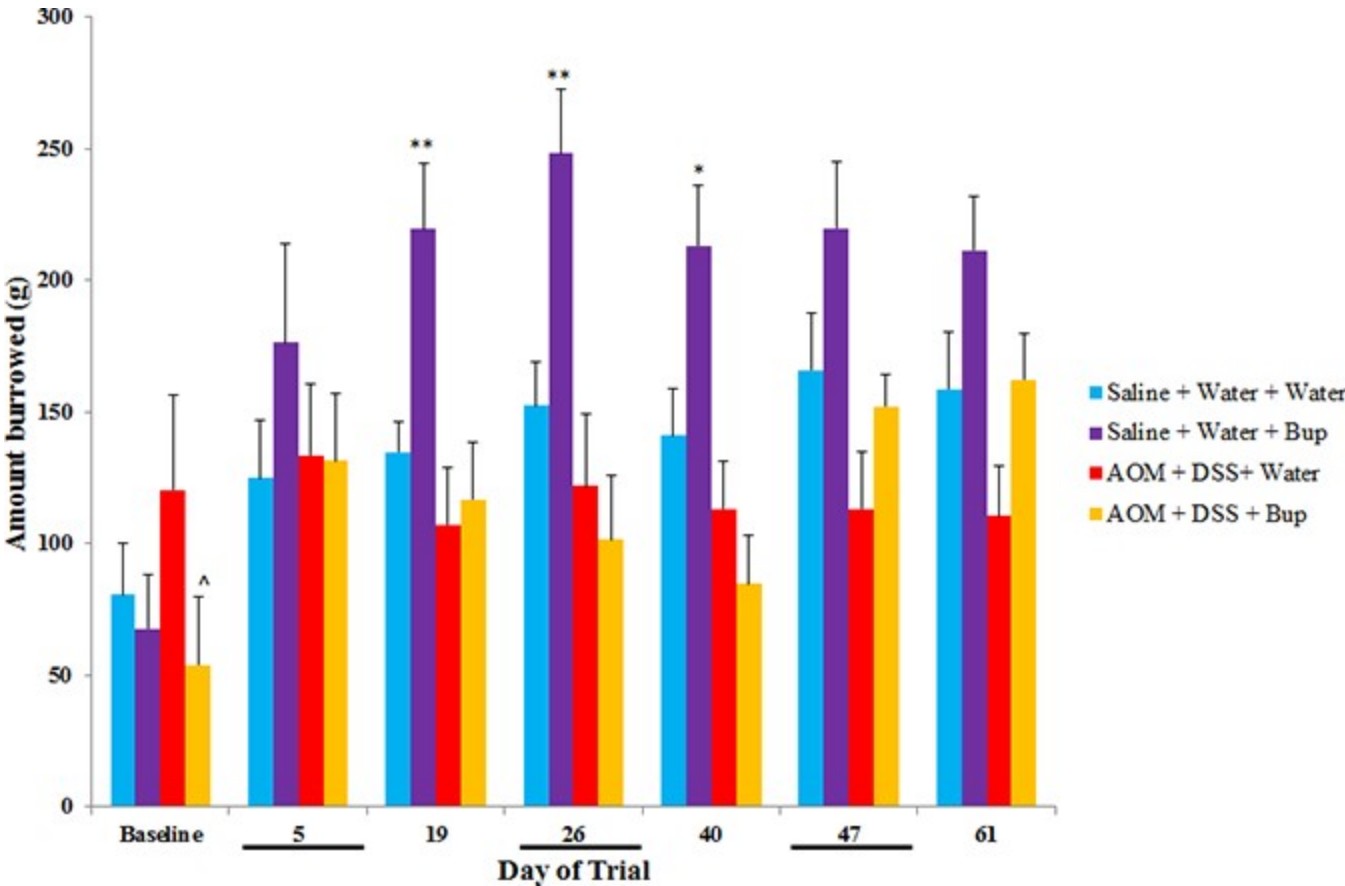

**Fig 5. Burrowing activity (n = 10/group).** Data were analysed using a repeated measures ANOVA with least significance difference (LSD) and are expressed as mean amount burrowed ± SEM (black line on the x axis represents a dextran sulphate sodium; DSS week). ***p<0.001, **p<0.01, *p<0.05 compared to Saline + Water + Water, ^^^p<0.001, ^^p<0.01, ^p<0.05 compared to AOM + DSS + Water.

impact of opioid analgesic (buprenorphine) intervention on pain reduction in the measures used at selected time-points, as highlighted by the minimal differences in grimace scores, burrowing behaviour and DAI in disease mice. This implied either that: 1) These animals were

**Table 1. Real-time and retrospective MGS scores (n = 10/group).**

|  | Saline + Water + Water | | Saline + Water + Bup | | AOM + DSS + Water | | AOM + DSS + Bup | |
|---|---|---|---|---|---|---|---|---|
|  | Retrospective | Real-time | Retrospective | Real-time | Retrospective | Real-time | Retrospective | Real-time |
| **Baseline** | 0 ± 0 | 0 ± 0 | 0.133 ± 0.07 | 0 ± 0 | 0 ± 0 | 0 ± 0 | 0 ± 0 | 0 ± 0 |
| **Day 5** | 0.167 ± 0.06 | 0 ± 0 | 0.233 ± 0.11 | 0.056 ± 0.04 | 0.296 ± 0.12 | 0 ± 0 | 0.185 ± 0.10 | 0 ± 0 |
| **Day 19** | 0.067 ± 0.04 | 0 ± 0 | 0.167 ± 0.13 | 0 ± 0 | 0.185 ± 0.08 | 0.198 ± 0.08**# | 0.444 ± 0.22 | 0.111 ± 0.05#^ |
| **Day 26** | 0.267 ± 0.10 | 0 ± 0 | 0 ± 0 | 0 ± 0 | 0.222 ± 0.10 | 0.062 ± 0.03 | 0.333 ± 0.18 | 0.049 ± 0.05 |
| **Day 40** | 0.167 ± 0.09 | 0 ± 0 | 0.100 ± 0.05 | 0 ± 0 | 0.111 ± 0.08 | 0 ± 0 | 0.037 ± 0.04 | 0 ± 0 |
| **Day 47** | 0.100 ± 0.07 | 0 ± 0 | 0.133 ± 0.07 | 0 ± 0 | 0.259 ± 0.19 | 0.012 ± 0.012 | 0.333 ± 0.14 | 0.086 ± 0.06^ |
| **Day 61** | 0.267 ± 0.12 | 0 ± 0 | 0.133 ± 0.05 | 0 ± 0 | 0.259 ± 0.11 | 0 ± 0 | 0.148 ± 0.11 | 0 ± 0 |

Data were analysed non-parametrically using a Friedman test and a Kruskal-Wallis with a Mann-Whitney *post-hoc* test and are expressed as mean MGS score ± SEM.

**p<0.01 compared to Saline + Water + Water at same time-point

#p<0.05 compared to baseline within a group

^p<0.05 compared to day 40 within a group.

**Table 2. Correlations between data sets of behavioural and clinical indicators on days 19, 40 and 61.**

| | | Colitis Severity day 19 | Tumour Number day 19 | Colitis Severity day 40 | Tumour Number day 40 | Colitis Severity day 61 | Tumour Number day 61 |
|---|---|---|---|---|---|---|---|
| **Real-time Grimace** | **Correlation Coefficient** | 0.517 | 0.668 | n.e | n.e | n.e | n.e |
| | **Significance (2-tailed)** | 0.001*** | 0.000*** | n.e | n.e | n.e | n.e |
| **Photo Grimace** | **Correlation Coefficient** | 0.240 | 0.289 | -0.147 | -0.194 | 0.080 | -0.027 |
| | **Significance (2-tailed)** | 0.146 | 0.078 | 0.379 | 0.244 | 0.633 | 0.872 |
| **Burrowing** | **Correlation Coefficient** | -0.478 | -0.405 | -0.393 | -0.408 | -0.538 | -0.266 |
| | **Significance (2-tailed)** | 0.002** | 0.012* | 0.015* | 0.011* | 0.000*** | 0.106 |
| **DAI** | **Correlation Coefficient** | 0.689 | 0.764 | 0.751 | 0.650 | 0.863 | 0.838 |
| | **Significance (2-tailed)** | 0.000*** | 0.000*** | 0.000*** | 0.000*** | 0.000*** | 0.000*** |

Data were analysed using a non-parametric spearman-rho test.

***p<0.001

**p<0.01 and

*p<0.05.

Note–no correlation coefficients could be derived between real-time grimace data and other measures on day 40 and 61 due to number of zero scores (n.e.–not estimable due to no variation in real-time grimace scores [all scores were 0]).

not experiencing pain, 2) The tests utilised were not sensitive enough to detect the type of pain experienced, or that 3) buprenorphine was ineffective in the face at the chosen time-points in mice with colitis-associated colorectal cancer.

Non-facial indicators of pain such as abdominal twitching, hunching, writhing, and belly press were identified in mice throughout the experimental period. Although these characteristics are not considered in facial grimace scores, they have been identified as validated pain associated behaviours that commonly occur following laboratory procedures [20, 21]. This highlights a key point in comparing real-time with retrospective measures especially when using personnel experienced with mice as real-time observers. Experienced observers are likely to subconsciously note general condition, and other pain-like behaviours such as hunching, writhing, belly press and immobility which may influence their scoring. These indicators are unable to be scored with a head-only photo image. Therefore, it would be advisable to use naïve observers for real-time grimace scoring in future studies.

The MGS action units have been validated in acute or moderate pain which lasts from minutes to hours. It has been reported that these action units are unable to be identified in mice days or weeks after a procedure, injury or surgical insult [5]. This is plausible since a fitness advantage would be gained by not communicating evidence of injury to predators via expression of the 'pain face' [22]. Consequently, the time-points selected in the current chronic study may have been too long after procedures to be able to identify pain present in the face. Furthermore, the MGS scores obtained were generally low (maximum 0.4 ± 0.2; live and retrospective analysis), implying a lack of sensitivity which may have precluded the finding of an analgesic effect. Similarly, in a study of rats with the gastrointestinal condition mucositis, Whittaker et al. (2016) reported that other behavioural measures utilised in the mucositis study including writhing, twitching, back-arching and sociability, to be more indicative of a pain status in the

disease rats than facial grimace [23]. Moreover, the low MGS results in the current study may have indicated that the mice were not experiencing pain, it could also highlight the evolutionary characteristic of mice hiding pain in their face to deter predators [4].

Animal ethics committees often recommend analgesic implementation in studies when animals are induced with disease, therefore, it is crucial to understand that analgesic intervention will not affect experimental design. In the current study, a minimal effect on colitis severity and DAI was observed in buprenorphine treated animals; however, these results were not consistently represented throughout the experimental period and may have been due to the timing of DAI scoring in respect to buprenorphine-administration. Furthermore, buprenorphine was orally-administered to mice thrice weekly for 63 days and it is possible that during this time mice established a tolerance to the analgesia. Dum and Herz (1981) concluded that rats subcutaneously injected with buprenorphine twice daily developed a tolerance after just five days [24]. Additionally, in a study of DBA/2 mice with SL2 lymphoma, there was no interaction between dietary-administered buprenorphine and time, indicating that a drug tolerance may have been established during the 20 day period [25]. Furthermore, Van Loo and authors (1997) concluded that there were no clear indicators that buprenorphine impacted the pain experienced by mice with tumours, concluding that it was an undesirable analgesia in a lymphoma tumour model [25]. Hence, these data cannot confirm an action of buprenorphine in reducing pain based on the MGS scores obtained, nor any improvement in wellbeing based on DAI score or burrowing behaviour. However, this needs to be considered in light of the difficulty in teasing apart beneficial, versus side effects using the DAI, and the differences obtained in baseline burrowing score. Moreover, buprenorphine does not modify experimental outcomes which is an important finding when considering analgesic use in gastrointestinal animal models.

In normal mice, burrowing activity was increased in buprenorphine–treated groups on days 19, 26 and 40. This hyper-excitability is supported by Cowan et al. (1977), whereby the authors documented that buprenorphine-administration increased activity (walking and hopping) in non-painful mice [12]. In another study, resting behaviours were decreased in buprenorphine-treated cancerous mice compared to controls [25]. Moreover, increased levels of activity are suggested to be a side-effect of buprenorphine administration in rodents [26]. In the current study, AOM/DSS control animals displayed a higher baseline (day -1) burrowing ability compared to AOM/DSS administered together with buprenorphine, which may have impacted the burrowing results obtained at the other time-points. In future studies, it would be beneficial to allocate treatment groups based on burrowing activity and adjust these to ensure that all experimental groups display similar burrowing abilities at baseline. Furthermore, there was no significant difference in burrowing activity between colitis-associated colorectal cancer and normal control mice in the current study, suggesting that burrowing is not an effective behavioural measure in the AOM/DSS model. Interestingly, DSS-administration alone has been reported to significantly impact burrowing behaviour in mice with acute [27] and chronic [28] colitis.

## Conclusions

Although the MGS scores obtained through real-time and retrospective analyses were unable to be validated in regards to pain alleviation in this chronic study, we were able to conclude that real-time grimace scores and daily clinical scores were correlated with increased colitis severity and tumour number across treatment groups. However, retrospective grimace scores were not correlated with other data sets. This indicated that real-time grimace may be a more accurate technique to complement other measures of disease in animal studies. However, as

previously discussed, there are limitations with the use of this method as a practical tool. Importantly, burrowing activity was negatively correlated with colitis severity and tumour number, indicating that as chronic disease develops, mouse behaviour will decrease as wellbeing is impacted. Given the lack of statistically-significant differences between groups we cannot recommend this measure in the colitis-associated colorectal cancer model. We conclude that the traditional disease activity index, or clinical score, presents the most comprehensive welfare assessment tool in the colitis-associated colorectal cancer mouse model, having a degree of sensitivity and comprising of both objective and subjective measures to constitute a final score.

In the current study, use of the MGS was unable to identify pain in the mouse model of colitis-associated colorectal cancer. Furthermore, the live-scoring MGS method was unable to be validated in this model of chronic gastrointestinal disease. Nonetheless, this study is the first to use the MGS in a chronic model of colorectal pain and it is the first to discuss the correlation between live and retrospective scoring methods with other study measurements. Further investigation of the MGS in this model is necessary to validate its reliability in chronic disease Consideration should be given to use of other methods for measuring ADL or affective state, for example nest making or judgement biasing in models of chronic disease. Moreover, DAI or clinical scores may be the most reliable method for affective state assessment in mouse models of chronic gastrointestinal diseases.

## Supporting information

**S1 Dataset. Disease activity index data files.**
(XLSX)

**S2 Dataset. Colonoscopically-Assessed colitis severity and tumour scores data files.**
(XLSX)

**S3 Dataset. Burrowing data files.**
(XLSX)

**S4 Dataset. Live mouse grimace scale data files.**
(XLSX)

**S5 Dataset. Retrospective mouse grimace scale data files.**
(XLSX)

**S6 Dataset. Correlation output data.**
(DOCX)

## Acknowledgments

The authors would like to acknowledge Rebecca George and Chloe Mitchell for assisting with real-time grimace scoring.

## Author Contributions

**Conceptualization:** Gordon S. Howarth, Alexandra L. Whittaker, Suzanne Mashtoub.

**Data curation:** Lauren C. Chartier, Michelle L. Hebart.

**Formal analysis:** Lauren C. Chartier, Michelle L. Hebart.

**Funding acquisition:** Lauren C. Chartier.

**Investigation:** Lauren C. Chartier.

**Methodology:** Lauren C. Chartier, Suzanne Mashtoub.

**Project administration:** Lauren C. Chartier.

**Resources:** Lauren C. Chartier.

**Supervision:** Gordon S. Howarth, Alexandra L. Whittaker, Suzanne Mashtoub.

**Validation:** Alexandra L. Whittaker.

**Visualization:** Lauren C. Chartier.

**Writing – original draft:** Lauren C. Chartier.

**Writing – review & editing:** Lauren C. Chartier, Gordon S. Howarth, Alexandra L. Whittaker, Suzanne Mashtoub.

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
