## [Decision Letter · Decision Letter 0]

18 Oct 2019

PONE-D-19-22786

Affective state determination in a mouse model of colitis-associated colorectal cancer

PLOS ONE

Dear Dr Mashtoub,

Thank you for submitting your manuscript to PLOS ONE. After careful consideration, we feel that it has merit but does not fully meet PLOS ONE’s publication criteria as it currently stands. Therefore, we invite you to submit a revised version of the manuscript that addresses the points raised during the review process.

We would appreciate receiving your revised manuscript within 3 months. To enhance the reproducibility of your results, we recommend that if applicable you deposit your laboratory protocols in protocols.io, where a protocol can be assigned its own identifier (DOI) such that it can be cited independently in the future. For instructions see: http://journals.plos.org/plosone/s/submission-guidelines#loc-laboratory-protocols

We look forward to receiving your revised manuscript.

Kind regards,

Mathilde Body-Malapel

Academic Editor

PLOS ONE

Journal Requirements:

"LCC received partial funding from The Australian Veterinary Association (Animal Welfare Trust Grantl; www.ava.com.au). The funders had no role in study design, data collection and analysis, decision to publish, or preparation of the manuscript."

a. Please provide an amended statement that declares *all* the funding or sources of support (whether external or internal to your organization) received during this study, as detailed online in our guide for authors at http://journals.plos.org/plosone/s/submit-now

Please also include the statement “There was no additional external funding received for this study.” in your updated Funding Statement.

Reviewers' comments:

Reviewer's Responses to Questions

**Comments to the Author**

1. Is the manuscript technically sound, and do the data support the conclusions?

Reviewer #1: Yes

Reviewer #2: Partly

2. Has the statistical analysis been performed appropriately and rigorously? 

Reviewer #1: Yes

Reviewer #2: I Don't Know

3. Have the authors made all data underlying the findings in their manuscript fully available?

Reviewer #1: Yes

Reviewer #2: No

4. Is the manuscript presented in an intelligible fashion and written in standard English?

Reviewer #1: Yes

Reviewer #2: Yes

5. Review Comments to the Author

Reviewer #1: In the present study, the AA evaluated behavioral assessment, mouse grimace score (MGS) and burrowing, as indicators of affective state in a mouse model of colitis-associated colorectal cancer. Furthermore, this study compares real-time MGS with retrospective MGS.

The purpose of the study is to implement and evaluate affective state in the disease model to better optimize analgesia protocols which is important for the welfare of laboratory animals.

Control groups receiving analgesia, buprenorphine, were used to assess any pain-associated effect on the used behavior tests (MGS and burrowing).

In the present study the AA show that burrowing and MSG (both real-time and retrospective) was not affected in the colitis-associated colorectal cancer models, while the traditional measures DAI score and colonoscopy severity was increased. No consistent effect of buprenorphine was present when comparison with control groups, which might be due to insensitiveness of the behavior test or simply that the mice did not experienced any pain.

The real-time and retrospective MSG correlated to each other, but only real-time MSG correlated with colitis severity at day 19.

The study concludes that behavioral testing with MGS and burrowing should not be used for pain assessment in this disease model where traditional methods like DAI and colonoscopy is still preferable.

The study is well-written although I think the general purpose of the study could be stated more clear in the introduction. Should these behavioral tests replace traditional measurements or be done in addition to?

Minor Points:

- Aim should definitely be stated more clearly in the discussion

- Specify DAI and colonoscopy parameters – what is the maximum possible score?

- Statistical method used should be stated in all figure legends

- Table 2: correlation coefficients from all real-time MSG points should be stated even though they do not correlate to colitis severity. All other coefficients is stated (also the not significant ones) for retrospective and burrowing.

- Organ weights and length. These results are not well integrated in the manuscript. The results are not evaluated in any figure or table. They are not discussed and they do not contribute to the conclusion.

- Figures and legends: Abbreviation should only be used when writing out in the figure legend (WCH, ARC, i.p, DSS, scope)

Reviewer #2: The manuscript titled “Affective state determination in a mouse model of colitis-associated colorectal cancer” by Chartier et al. investigated whether the Mouse Grimace Scale and the burrowing test are a useful tool to assess the well-being of mice with colitis-associated colorectal cancer when compared to disease activity index. The results of the study indicated that neither of these two methods reliably measured the well-being while the clinical scores were increased in diseased mice and were positively correlated with colonoscopically-assessed severity and tumor number. The content of the manuscript is of interest for researchers using the AOM/DSS mouse model of colitis-associated colorectal cancer as well as laboratory animal veterinarians and can contribute to a better understanding of welfare indicators that are sensible to detect compromised well-being in this animal model. Overall, the manuscript is well-written, it is clear and nicely organized.

Issues that should be addressed in the revision of the manuscript:

- The half-life of buprenorphine is quite short (lasting effects: approx. 6-8 hours) and therefore it is currently discussed to use sustained release formulation of buprenorphine providing consistent, long-lasting analgesia. I was wondering how the authors made sure that the serum concentrations of buprenorphine were consistent until mice received the next treatment. Did the effect of a single dosage last for 3 days? It would be important to know the exact interval (in hours) between the last buprenorphine treatment and the analysis of MGS, burrowing tests, and clinical scoring.

- In the present study, water or buprenorphine were administered by oral gavage although there are several refinement methods for this procedure that prevent the mice from the distress caused by oral gavage. Buprenorphine can be administered via the drinking water or using flavored gelatin or Nutella etc. I would appreciate if you could explain in the section "experimental design" why these refinement methods were not applied.

- Data availability: The authors state that all relevant data are within the manuscript and its supporting information files. Since I have not received the supporting information files, I cannot check whether this statement is true.

- P5L95 Introduction: Miller and Leach also compared live versus retrospective MGS scores and found that in general live scores were lower than those obtained from images, that contradicts the results of the present study. Miller and Leach did not use the same mouse model as the authors of the present study. However, I would like to encourage the authors to discuss the discrepancy in their MGS results. Miller, Amy L., and Matthew C. Leach. "The mouse grimace scale: a clinically useful tool?." PloS one 10.9 (2015): e0136000.

- P5-6L104-114 Animal studies: Please explain why female mice were used only and provide more details of relative humidity, weights of animals, cage enrichment, type of bedding material, and number of cage companions. Explain how the number of animals was arrived at and provide details of sample size calculation used. How were animals allocated to experimental groups (details of randomization).

- P6L120-125: Please indicate the administration volumes for water, buprenorphine, saline, and AOM, injection site and size of cannula used for the ip injection, and the route of administration for DSS/water (L123-124).

- P7LL136-141 DAI: Were the experimenters blinded when scoring the parameters?

- P7L143-153 Colonoscopy: How long did the procedure last? Provide details of the compound isoflurane (supplier etc.).

- P8L155-163: Was the same burrow used as described by Deacon? Please add more information on the pebbles used for this test (approximate size, supplier etc.) and the time of day when the test was conducted.

- P8L165ff MGS: How many persons generated live and retrospective scores? It is advisable that more than one scorer is deployed in MGS scoring and the interrater reliability is calculated. Could you provide more details of the experience of experimenters in using the MGS and the time of day when live MGS scores were obtained?

- P9LL174-180: I cannot follow the explanation of calculation. Could you provide an example calculation.

- P9-10 Statistical analysis: Did you check for normal distribution of data?

- P10-18 Results: Provide details of the statistical methods used for each analysis and test statistics (not p value only).

- P12-13L251ff MGS: Did you examine whether live and retrospective MGS score correlate?

- P18L295 Discussion: While the MGS was originally developed to assess pain, it is currently discussed that changes in facial expression described in the MGS are not present in pain only but also in other affective states. Moreover, the weights of the facial action units seem to vary between the different states (Langford et al. 2010, . Could the authors determine which of the facial features were most affected? I would kindly ask the authors to adapt their statement that the MGS is specific to pain (it is not).

Dalla Costa, Emanuela, et al. "Can grimace scales estimate the pain status in horses and mice? A statistical approach to identify a classifier." PloS one 13.8 (2018): e0200339.

Langford, Dale J., et al. "Coding of facial expressions of pain in the laboratory mouse." Nature methods 7.6 (2010): 447.

- P19L309-311 Discussion: Depending on the interval between the last administration of buprenorphine and the assessment of well-being using the MGS or the burrowing test, the three hypotheses should be rethought. With regard to hypothesis #2) the authors should also consider the issues raised above (number of MGS scorers, experience of MGS scorers)

- P20L322: I see your point in using naïve scorers not being familiar with mouse behavior, though I think it is crucial that they are properly trained in using the MGS.

- P22L373: How do you explain the discrepancy between your findings and results of Safaeian et al.?

6. PLOS authors have the option to publish the peer review history of their article (what does this mean?). If published, this will include your full peer review and any attached files.

Reviewer #1: No

Reviewer #2: No

---

## [Author Response · Author response to Decision Letter 0]

12 Nov 2019

Responses to Reviewers

Title: Affective state determination in a mouse model of colitis-associated colorectal cancer

Authors: Lauren C Chartier, Michelle L Hebart, Gordon S Howarth, Alexandra L Whittaker and Suzanne Mashtoub

Manuscript ID: PONE-D-19-22786

The authors would kindly like to thank the two reviewers for their comments and questions regarding the current manuscript. All answers to comments and revisions have been documented in the manuscript file labelled ‘Revised Manuscript with Track Changes. Page/line number references for amendments are listed on this document for your convenience. Additionally, a copy of the completed reviewed manuscript has been uploaded with the revision and is labelled ‘Manuscript’. 

Reviewer #1: In the present study, the AA evaluated behavioural assessment, mouse grimace score (MGS) and burrowing, as indicators of affective state in a mouse model of colitis-associated colorectal cancer. Furthermore, this study compares real-time MGS with retrospective MGS. The purpose of the study is to implement and evaluate affective state in the disease model to better optimize analgesia protocols which is important for the welfare of laboratory animals. Control groups receiving analgesia, buprenorphine, were used to assess any pain-associated effect on the used behaviour tests (MGS and burrowing). In the present study the AA show that burrowing and MSG (both real-time and retrospective) was not affected in the colitis-associated colorectal cancer models, while the traditional measures DAI score and colonoscopy severity was increased. No consistent effect of buprenorphine was present when comparison with control groups, which might be due to insensitiveness of the behaviour test or simply that the mice did not experienced any pain. The real-time and retrospective MSG correlated to each other, but only real-time MSG correlated with colitis severity at day 19. The study concludes that behavioural testing with MGS and burrowing should not be used for pain assessment in this disease model where traditional methods like DAI and colonoscopy is still preferable. The study is well-written although I think the general purpose of the study could be stated more clear in the introduction. Should these behavioural tests replace traditional measurements or be done in addition to?

Minor Points:

- Aim should definitely be stated more clearly in the introduction

The final paragraph of the introduction has been amended to ensure that the aim of the study is clear. We aimed to assess all behavioural/pain assessments (clinical scoring, MGS and burrowing) in order to identify the most reliable method for pre-clinical models of colitis-associated colorectal cancer. Furthermore, pain assessment using the real/retrospective MGS had not yet been validated in a model of colitis-associated colorectal cancer prior to this manuscript (P5L92-95). 

- Specify DAI and colonoscopy parameters – what is the maximum possible score?

We have included the maximum possible scores for these parameters in the methods section of the manuscript. The maximum score for DAI is 12, and for colonoscopically-assessed severity is 15.

- Statistical method used should be stated in all figure legends

We have amended and included the statistical method used in all figure legends. 

- Table 2: correlation coefficients from all real-time MSG points should be stated even though they do not correlate to colitis severity. All other coefficients is stated (also the not significant ones) for retrospective and burrowing.

Correlation coefficients for real-time MGS were originally not stated in Table 2 for days 40 and 61 as the correlation coefficient cannot be calculated due to the fact that real-time grimace scores were all zero for these time-points. We have amended Table 2 and placed not estimable (n.e) in these time-points and a note in the legend to clarify that there was no variation in grimace scores on days 40 and 61. 

- Organ weights and length. These results are not well integrated in the manuscript. The results are not evaluated in any figure or table. They are not discussed and they do not contribute to the conclusion.

We agree with your observation and have thus removed the ‘Organ weights and length’ data and section from the manuscript.

- Figures and legends: Abbreviation should only be used when writing out in the figure legend (WCH, ARC, i.p, DSS, scope)

All abbreviations have been stated in full prior to being abbreviated throughout the manuscript. Moreover, figure legends have been amended to include the full word before being abbreviated (e.g. dextran sulphate sodium; DSS). 

 

Reviewer #2: The manuscript titled “Affective state determination in a mouse model of colitis-associated colorectal cancer” by Chartier et al. investigated whether the Mouse Grimace Scale and the burrowing test are a useful tool to assess the well-being of mice with colitis-associated colorectal cancer when compared to disease activity index. The results of the study indicated that neither of these two methods reliably measured the well-being while the clinical scores were increased in diseased mice and were positively correlated with colonoscopically-assessed severity and tumor number. The content of the manuscript is of interest for researchers using the AOM/DSS mouse model of colitis-associated colorectal cancer as well as laboratory animal veterinarians and can contribute to a better understanding of welfare indicators that are sensible to detect compromised well-being in this animal model. Overall, the manuscript is well-written, it is clear and nicely organized.

Issues that should be addressed in the revision of the manuscript:

- The half-life of buprenorphine is quite short (lasting effects: approx. 6-8 hours) and therefore it is currently discussed to use sustained release formulation of buprenorphine providing consistent, long-lasting analgesia. I was wondering how the authors made sure that the serum concentrations of buprenorphine were consistent until mice received the next treatment. Did the effect of a single dosage last for 3 days? It would be important to know the exact interval (in hours) between the last buprenorphine treatment and the analysis of MGS, burrowing tests, and clinical scoring.

Mice were administered thrice weekly with buprenorphine. These administrations occurred in the morning, 2 hours prior to MGS scoring and 8 hours prior to burrowing analyses. Due to the timings of these analyses we expected that we would be able to observe the effect of buprenorphine as the analyses occurred within the estimated half-life. However, as clinical scoring was measured first thing in the morning for initial monitoring of all mice, these scores were obtained prior to buprenorphine administration. As a result we understand that we may not have picked up the effect of buprenorphine in the clinical scoring analyses. We have now indicated specific time-frames following buprenorphine administration for each behavioural test in the methods section of the manuscript. 

- In the present study, water or buprenorphine were administered by oral gavage although there are several refinement methods for this procedure that prevent the mice from the distress caused by oral gavage. Buprenorphine can be administered via the drinking water or using flavored gelatin or Nutella etc. I would appreciate if you could explain in the section "experimental design" why these refinement methods were not applied.

Buprenorphine was administered via oral gavage as the control groups utilised in this study were also used as part of another study and thus were receiving water oral gavages. Although we acknowledge that some minor distress is caused through the gavage procedure, we administered buprenorphine in this manner to ensure that all mice were exposed to the same procedures and therefore significant results (e.g body weight loss) could not be attributed to the procedures alone. We have included an extra sentence justifying the administration route of buprenorphine in the experimental design section (P7L133-134).

- Data availability: The authors state that all relevant data are within the manuscript and its supporting information files. Since I have not received the supporting information files, I cannot check whether this statement is true.

All supporting raw data files have been uploaded with the revised versions of this manuscript on Figshare. 

- P5-6L104-114 Animal studies: Please explain why female mice were used only and provide more details of relative humidity, weights of animals, cage enrichment, type of bedding material, and number of cage companions. Explain how the number of animals was arrived at and provide details of sample size calculation used. How were animals allocated to experimental groups (details of randomization).

Further details on housing humidity, average animal weight at the beginning of the experiment and enrichment items have now been included in the Animal studies section of the manuscript (P5P6L105-122). Only female mice were utilised in this study to remain consistent with data obtained from previous trials using the AOM/DSS model. A statement to highlight this reasoning has been included in the ‘Animal studies’ section of the manuscript with appropriate referencing. However, we understand that an initial study with male mice needs to be investigated in the future. Mice were stratified to groups based on day 0 body weight and baseline burrowing activity to ensure a spread of weight and burrowing ability across all treatment groups. Group size was calculated using Clin.Calc for mouse grimace scale outcomes from Rosen et al. (2017). This calculation assumed a power of 80%, and indicated that a minimum sample size of n=9/group was necessary. Therefore, as we included a n=10/group we ensured that statistical power would be reached. The details of this power calculation have now also been included in the Animal studies/ experimental design section of the manuscript for clarification (P6L124-138). 

- P6L120-125: Please indicate the administration volumes for water, buprenorphine, saline, and AOM, injection site and size of cannula used for the ip injection, and the route of administration for DSS/water (L123-124).

The oral-gavage administration volumes of water and buprenorphine are already stated in the manuscript as 80µL (P6L123). The volumes of saline and AOM injections were calculated based on individual mouse body weight (7.4mg/kg; P6L125). However, for clarification we have included an average injection volume (0.14mL) in this section of the manuscript, along with the needle gauge used (27G) for the intraperitoneal injections. DSS/water was provided in drinking bottles for ad libitum access, these details have also been included in the manuscript (P6L127). 

- P7LL136-141 DAI: Were the experimenters blinded when scoring the parameters?

DAI scoring was not performed in a blinded-fashion as it was measured routinely by the researchers when conducting initial monitoring of all mice first thing in the morning. This is because DAI/ clinical scoring is required for ethical purposes and is crucial for determining humane end-points. Furthermore, the euthanasia criteria accepted by the animal ethics committees that approved this study is based upon the DAI scoring system, hence it was important to provide un-biased scores. Moreover, the researchers that monitor and obtain DAI scores in this study are experienced and adequately trained to provide standardised scores for each parameter. 

- P7L143-153 Colonoscopy: How long did the procedure last? Provide details of the compound isoflurane (supplier etc.).

Isoflurane was sourced from AbbVie Inc., and the colonoscopy procedure lasted approximately 10 minutes from anaesthetic induction to recovery. This has now been detailed in the revised manuscript (P8L156-167).

- P8L155-163: Was the same burrow used as described by Deacon? Please add more information on the pebbles used for this test (approximate size, supplier etc.) and the time of day when the test was conducted.

The burrows used in the current study were modified from those detailed by Deacon. We used Coca-Cola bottles (diameter and length provided) and kitty litter as pebbles. These burrowing analyses were conducted at 6pm, one hour after commencement of the dark cycle and 8 hours following buprenorphine administration. These specific details have been included in the revised manuscript (P8L170-179). 

- P8L165ff MGS: How many persons generated live and retrospective scores? It is advisable that more than one scorer is deployed in MGS scoring and the interrater reliability is calculated. Could you provide more details of the experience of experimenters in using the MGS and the time of day when live MGS scores were obtained?

Live and retrospective MGS scores were obtained from one experiment-blinded observer at each time-point throughout. Although it is advisable to have more than one scorer, previous studies have also been published with one observed (Cho C, Michailidis, V., Lecker, I. et al. Evaluating analgesic efficacy and administration route following craniotomy in mice using the grimace scale. Sci Rep 9, 359 (2019); Akintola, T., Raver C, Studlack P. et al. The grimace scale reliably assesses chronic pain in a rodent model of trigeminal neuropathic pain. Neurobiol Pain 2, 13-17 (2017). Furthermore, live MGS scores/ videos for retrospective scoring were obtained in the morning (9-12pm; 2 hours following buprenorphine administration), as detailed in the manuscript (P9L200). The scorers conducting were experienced in grimace identification but naïve to experimental conditions. Dr Alexandra Whittaker and Miss Rebecca George have published studies utilising the grimace scale (George RP, Howarth GS, Whittaker AL. Use of the Rat Grimace Scale to Evaluate Visceral Pain in a Model of Chemotherapy-Induced Mucositis. Animals (Basel). 2019 Sep 12;9(9); Whittaker AL, Leach MC, Preston FL, Lymn KA, Howarth GS. Effects of acute chemotherapy-induced mucositis on spontaneous behaviour and the grimace scale in laboratory rats. Lab Anim. 2016 Apr;50(2):108-18.) and Miss Lauren Chartier has performed grimace analyses in two studies (unpublished). Furthermore, training in the scoring techniques was performed using the grimace posters provided in the publication by Langford et al. (2010) and using old video data. 

- P9LL174-180: I cannot follow the explanation of calculation. Could you provide an example calculation?

We have edited the written explanation to be more clear of the real-time MGS score calculation (P9L214-228) and have included a table with a sample calculation below. 

Time Orbital tightening Ear positon Whisker change Cheek bulge Nose bulge Median per 15 sec Average per 90 sec interval

15-30sec 1 0 1 2 1 1 0.333333333

45-1:00 0 0 0 0 0 0 

1:15-1:30 1 0 0 0 0 0 

1:45-2:00 0 1 1 2 1 1 1

2:15-2:30 1 1 1 1 1 1 

2:45-3:00 0 2 1 1 1 1 

3:15-3:30 0 2 1 0 0 0 0.25

3:45-4:00 0 0 0 0 0 0 

4:15-4:30 1 0 0 0 0 0 

4:45-5:00 0 1 1 2 1 1 

 Final Score (mean of score per 90 sec) 0.527777778

- P9-10 Statistical analysis: Did you check for normal distribution of data?

Yes, as mentioned in the statistical analysis section of the manuscript, data was tested for normality using a Shapiro–Wilk test and then analysed accordingly (P10L199). Furthermore, a Levene’s test for homogeneity of variance was used and no outliers were identified. 

- P10-18 Results: Provide details of the statistical methods used for each analysis and test statistics (not p value only).

The details of the statistical method used for each analysis is highlighted in the statistical analysis section of the manuscript. Additionally, these details have been included in the figure legends. 

- P12-13L251ff MGS: Did you examine whether live and retrospective MGS score correlate?

All data from real-time and retrospective MGS scores had a correlation coefficient of 0.064, and thus are not strongly correlated. Furthermore, at all other time-points, the correlation coefficient for real-time and retrospective MGS are as follows; -0.017 (day 5), 0.163 (day 19), -0.05 (day 26), not estimable (day 40), 0,156 (day 47) and not estimable (day 61). 

- P18L295 Discussion: While the MGS was originally developed to assess pain, it is currently discussed that changes in facial expression described in the MGS are not present in pain only but also in other affective states. Moreover, the weights of the facial action units seem to vary between the different states (Langford et al. 2010, . Could the authors determine which of the facial features were most affected? I would kindly ask the authors to adapt their statement that the MGS is specific to pain (it is not).

In the current study ‘orbital tightening’ and ‘ear position’ were the features of the MGS that were most affected. Other features such as ‘whisker change’ can be quite difficult to score on C57BL/6 mice as their dark coat makes it difficult to distinguish the whiskers in photographs. Moreover, we have amended the statement in the discussion as per this comment that the MGS is not specific to pain (P19L330).

- P19L309-311 Discussion: Depending on the interval between the last administration of buprenorphine and the assessment of well-being using the MGS or the burrowing test, the three hypotheses should be rethought. With regard to hypothesis #2) the authors should also consider the issues raised above (number of MGS scorers, experience of MGS scorers)

As mentioned in previous responses to comments and in the manuscript, MGS scoring occurred 2 hours following buprenorphine administration, and burrowing analyses at 8 hours post buprenorphine administration. The number of MGS scorers and relative experience has been clarified in the manuscript and in an above comment to the reviewer responses, therefore, no further issues have been raised in regards to hypothesis (2) The tests utilised were not sensitive enough to detect the type of pain experienced. 

- P20L322: I see your point in using naïve scorers not being familiar with mouse behavior, though I think it is crucial that they are properly trained in using the MGS.

We agree that proper training in the MGS criteria and features are necessary for reliable results, however, as mentioned in the discussion, experienced observers may subconsciously score based on other bodily measures of pain identified during live scoring. Future investigations could compare naïve scorers of live MGS and trained/experienced observers for retrospective MGS is used, as experienced observes will not be able to score non-facial measures of pain if only using facial photographs. 

- P22L373: How do you explain the discrepancy between your findings and results of Safaeian et al.?

Safaeian et al. reported that DSS-treated mice presented with significantly reduced burrowing activity compared to normal controls. However, these results were founded in a model of chronic colitis, not colitis-associated colorectal cancer. Although the experimental models for chronic colitis and colitis-associated colorectal cancer share some similarities, the time-points at which burrowing was assessed in the Safaeian study were different to that used in the current study, as the timelines differ. Therefore, although we would hypothesise that the impact of burrowing ability would be similar in both disease models, the varying time-points for the measurement of burrowing between the studies is potentially what accounts for the discrepancy in our outcomes.

---

## [Decision Letter · Decision Letter 1]

5 Dec 2019

PONE-D-19-22786R1

Affective state determination in a mouse model of colitis-associated colorectal cancer

PLOS ONE

Dear Dr Mashtoub,

Thank you for submitting your manuscript to PLOS ONE and for revising your manuscript. It has been substantially improved but there are still a few issues that should be addressed. Therefore, we invite you to submit a revised version of the manuscript that addresses the few points raised during the second review process.

We would appreciate receiving your revised manuscript within 2 months. To enhance the reproducibility of your results, we recommend that if applicable you deposit your laboratory protocols in protocols.io, where a protocol can be assigned its own identifier (DOI) such that it can be cited independently in the future. For instructions see: http://journals.plos.org/plosone/s/submission-guidelines#loc-laboratory-protocols

We look forward to receiving your revised manuscript.

Kind regards,

Mathilde Body-Malapel

Academic Editor

PLOS ONE

Journal Requirements:

Reviewers' comments:

Reviewer's Responses to Questions

**Comments to the Author**

1. If the authors have adequately addressed your comments raised in a previous round of review and you feel that this manuscript is now acceptable for publication, you may indicate that here to bypass the “Comments to the Author” section, enter your conflict of interest statement in the “Confidential to Editor” section, and submit your "Accept" recommendation.

Reviewer #1: All comments have been addressed

Reviewer #2: (No Response)

2. Is the manuscript technically sound, and do the data support the conclusions?

Reviewer #1: Yes

Reviewer #2: Partly

3. Has the statistical analysis been performed appropriately and rigorously? 

Reviewer #1: Yes

Reviewer #2: N/A

4. Have the authors made all data underlying the findings in their manuscript fully available?

Reviewer #1: Yes

Reviewer #2: Yes

5. Is the manuscript presented in an intelligible fashion and written in standard English?

Reviewer #1: Yes

Reviewer #2: Yes

6. Review Comments to the Author

Reviewer #1: All comments have been satisfactory and thoroughly addressed. I recommend the current manuscript for publication in PLOS ONE.

Reviewer #2: I thank the authors for their response to my questions and comments. However, there are still a few points that have not been addressed and revised in the manuscript:

- Although you mentioned that “mice were stratified to groups based on baseline body weight and burrowing activity” (P6LL128-130), there were significant differences in burrowing activity at baseline and you stated in the discussion part that “in future studies, it would be beneficial to allocate treatment groups based on the burrowing activity”. Please check the statement given in the part “experimental design” in LL128-130.

- Please provide information on the health status of the mice (see ARRIVE guidelines).

- Material and Methods, DAI (P8LL157-162): Please indicate that scorer(s) were not blinded and that DAI scores were obtained before administration of buprenorphine.

- Material and Methods, MGS: Time of day was not added to this section. Moreover, the authors described that there was one MGS scorer only, but in the response to my comments three scorers are listed (Alexandra Whittaker, Rebecca George, and Lauren Chartier). Did each of them score a third of the mouse faces? If this is true please add this information to the MGS methods section.

- The authors stated in their responses: “Mice were administered thrice weekly with buprenorphine. These administrations occurred in the morning, 2 hours prior to MGS scoring and 8 hours prior to burrowing analyses.”, “Furthermore, live MGS scores/videos for retrospective scoring were obtained in the morning (9-12pm; 2 hours following

buprenorphine administration)”

I assume that PM is a typing error (AM?) and DAI was obtained at 9 AM, buprenorphine was administered at 9 AM, mouse faces were scored at noon (MGS) and burrowing was monitored at 6 PM. According to this schedule, DAI could not asses the analgesic effects of buprenorphine (in contrast to the MGS and the burrowing test). Therefore, I would recommend emphasizing this issue and make it clearer for the reader in the discussion (especially in P20LL335-346; in this passage the reader has the impression that DAI was considered to assess the analgesic effect of buprenorphine administered on the respective days). Due to the short lasting analgesic effects of buprenorphine, it is unlikely that you have “picked up the effect of buprenorphine in the clinical scoring analysis”, as you also stated in your response, but of course you may have measured side effects.

Moreover, results of the burrowing test should be interpreted with more caution in the discussion part as baseline scores were significantly different – this makes it very tricky and should be pointed out for the reader.

The statement “Overall, results were unable to conclude a significant impact of opioid analgesic (buprenorphine) intervention on the measures used, highlighted by the minimal differences in grimace scores, burrowing behaviour and DAI in disease mice.” (P20LL340-343) does not consider, 1) that DAI also includes pain-specific behaviors and was actually increased in AOM+DSS groups, 2) that the time of DAI scoring did not allow for the assessment of analgesic effects of buprenorphine, 3) and that burrowing data are difficult to interpret due to baseline differences.

Based on the difficulties stated above, I would also recommend rethinking the three hypotheses given in LL343-346.

Moreover, the following phrase needs to be reformulated with regard to the above-mentioned concerns: “Hence, taken together the results suggest that buprenorphine is ineffective in improving wellbeing in mice with colitis associated colorectal cancer.” (P22LL388-390)

7. PLOS authors have the option to publish the peer review history of their article (what does this mean?). If published, this will include your full peer review and any attached files.

Reviewer #1: No

Reviewer #2: No

---

## [Author Response · Author response to Decision Letter 1]

12 Jan 2020

Responses to Reviewers – 13th January 2020

Title: Affective state determination in a mouse model of colitis-associated colorectal cancer

Authors: Lauren C Chartier, Michelle L Hebart, Gordon S Howarth, Alexandra L Whittaker and Suzanne Mashtoub

Manuscript ID: PONE-D-19-22786

The authors would kindly like to thank the two reviewers for their comments and questions regarding the current manuscript. All answers to comments and revisions have been documented in the manuscript file labelled ‘Revised Manuscript with Track Changes’. Page/line number references for amendments are listed in this document for your convenience. Additionally, a copy of the completed reviewed manuscript has been uploaded with the revision and is labelled ‘Manuscript’. 

Reviewer #1: All comments have been satisfactory and thoroughly addressed. I recommend the current manuscript for publication in PLOS ONE.

Reviewer #2: I thank the authors for their response to my questions and comments. However, there are still a few points that have not been addressed and revised in the manuscript:

- Although you mentioned that “mice were stratified to groups based on baseline body weight and burrowing activity” (P6LL128-130), there were significant differences in burrowing activity at baseline and you stated in the discussion part that “in future studies, it would be beneficial to allocate treatment groups based on the burrowing activity”. Please check the statement given in the part “experimental design” in LL128-130.

It was originally stated that bodyweight and burrowing ability was used to assign groups as mice were firstly stratified to treatment groups based on their baseline bodyweight, then baseline burrowing data was used to ensure that burrowing ability was equally distributed across groups. The significant differences in baseline burrowing ability was unavoidable due to control groups being utilised in another study of the same nature running at the same time, as stated in the ‘animal studies’ section of the Material and Methods. However, to avoid confusion for the reader, the authors have revised the ‘experimental design’ section and now reads ‘Mice were stratified to groups based on baseline body weight.’ (P6L129). The statement in the discussion remains as “in future studies, it would be beneficial to allocate treatment groups based on the burrowing activity”.

- Please provide information on the health status of the mice (see ARRIVE guidelines).

The health status of mice has now been included in the ‘Animal Studies’ section of the Methods. The included statement (P6L117-119) reads; ‘The ARC undertakes a quarterly health screening, covering various bacterial, viral and parasitic organisms, all of which the obtained colony screened negative for.’

- Material and Methods, DAI (P8LL157-162): Please indicate that scorer(s) were not blinded and that DAI scores were obtained before administration of buprenorphine.

We have now included that scorers were not blinded to treatment groups when obtaining DAI measurements and that DAI was calculated prior to buprenorphine administration (P7L153-P8L160). 

- Material and Methods, MGS: Time of day was not added to this section. Moreover, the authors described that there was one MGS scorer only, but in the response to my comments three scorers are listed (Alexandra Whittaker, Rebecca George, and Lauren Chartier). Did each of them score a third of the mouse faces? If this is true please add this information to the MGS methods section.

In the MGS section of the experimental design, we stated that MGS scoring occurred in the morning following buprenorphine administration; however, to clarify this we have now included a time (approximately 9am-12pm) on P9L215. There was only one MGS scorer per mouse, however, due to the large number of animals and time restraints, live scoring was shared amongst the three grimace experienced researchers (Alexandra Whittaker, Rebecca George, and Lauren Chartier), whereby one researcher scored one mouse. Retrospective blinded scoring was all performed by Miss Lauren Chartier. 

- The authors stated in their responses: “Mice were administered thrice weekly with buprenorphine. These administrations occurred in the morning, 2 hours prior to MGS scoring and 8 hours prior to burrowing analyses.”, “Furthermore, live MGS scores/videos for retrospective scoring were obtained in the morning (9-12pm; 2 hours following

buprenorphine administration)”. I assume that PM is a typing error (AM?) and DAI was obtained at 9 AM, buprenorphine was administered at 9 AM, mouse faces were scored at noon (MGS) and burrowing was monitored at 6 PM. According to this schedule, DAI could not asses the analgesic effects of buprenorphine (in contrast to the MGS and the burrowing test). Therefore, I would recommend emphasizing this issue and make it clearer for the reader in the discussion (especially in P20LL335-346; in this passage the reader has the impression that DAI was considered to assess the analgesic effect of buprenorphine administered on the respective days). Due to the short lasting analgesic effects of buprenorphine, it is unlikely that you have “picked up the effect of buprenorphine in the clinical scoring analysis”, as you also stated in your response, but of course you may have measured side effects. 

 To clarify the timeline of events for analyses for all mice - DAI was measured first thing in the morning (7-8am) prior to buprenorphine administration (9am). Then MGS scoring/video recordings occurred between 9am-12pm, and finally burrowing analyses were performed from 6pm on selected days. We have included the approximate time of each analysis in the ‘Materials and Methods’ section of the manuscript as well as time following buprenorphine administration to avoid confusion. It is correct that at this timeline we were unlikely to pick up the analgesic effect of buprenorphine in DAI as the effects would not have lasted 24 hours, however this has been stated in the methods that DAI was measured prior to administration (P7L156). Furthermore, MGS was performed at a time where the effects of buprenorphine should have been measureable. 

Moreover, results of the burrowing test should be interpreted with more caution in the discussion part as baseline scores were significantly different – this makes it very tricky and should be pointed out for the reader.

 We have now added to the final paragraph of the discussion (P22L387-388) to highlight that burrowing was significantly different at baseline in AOM/DSS controls and therefore may have impacted the results obtained at other time-points within the study. 

The statement “Overall, results were unable to conclude a significant impact of opioid analgesic (buprenorphine) intervention on the measures used, highlighted by the minimal differences in grimace scores, burrowing behaviour and DAI in disease mice.” (P20LL340-343) does not consider, 1) that DAI also includes pain-specific behaviors and was actually increased in AOM+DSS groups, 2) that the time of DAI scoring did not allow for the assessment of analgesic effects of buprenorphine, 3) and that burrowing data are difficult to interpret due to baseline differences. Based on the difficulties stated above, I would also recommend rethinking the three hypotheses given in LL343-346.

1) P20L326 in the discussion section has been reworded to highlight that buprenorphine did not have an effect in terms of reducing pain (as this was the purpose of administering analgesia) in AOM/DSS mice, as buprenorphine did impact DAI scores at some time-points. 

2) In the current study, DAI was measured daily in the morning prior to buprenorphine administration. This time-point was selected as DAI was routinely incorporated into the daily monitoring of all mice in the study. Furthermore, the effect of buprenorphine on DAI parameters (including bodyweight loss and diarrhoea) did not confound the study as these are retrospective and slow pathological changes (P22L380-382). For example, if DAI was measured 30 mins follow buprenorphine administration, changes in bodyweight and stool consistency would not be identifiable in this short time-frame. 

3) The authors acknowledge that the discrepancies in baseline burrowing data mean that these results are difficult to interpret. This issue has been identified in the discussion (P22L390-392) and that future studies should aim to have groups with relatively similar baseline burrowing ability where possible. However, based on the burrowing results from other time-points in the study it is likely that buprenorphine does not have an effect on burrowing behaviour. 

In light of these responses, the authors still believe that the hypothesis mentioned on P19L328-331 of the discussion are reasonable. 

Moreover, the following phrase needs to be reformulated with regard to the above-mentioned concerns: “Hence, taken together the results suggest that buprenorphine is ineffective in improving wellbeing in mice with colitis associated colorectal cancer.” (P22LL388-390)

We agree that the statement highlighted is a little too strong and have modified it. The statement now reads ‘Hence, these data cannot confirm an action of buprenorphine in reducing pain based on the MGS scores obtained, nor any improvement in wellbeing based on DAI score or burrowing behaviour. However, this needs to be considered in light of the difficulty in teasing apart beneficial, versus side effects using the DAI, and the differences obtained in baseline burrowing score. (P21L378-383). 

Furthermore, earlier in the same paragraph we have added more clarification that the timing of DAI measurements in respect to buprenorphine-administration is likely why minimal effects were observed (P21L367-368).

---

## [Editor Report · Decision Letter 2]

15 Jan 2020

Affective state determination in a mouse model of colitis-associated colorectal cancer

PONE-D-19-22786R2

Dear Dr. Mashtoub,

We are pleased to inform you that your manuscript has been judged scientifically suitable for publication and will be formally accepted for publication once it complies with all outstanding technical requirements.

With kind regards,

Mathilde Body-Malapel

Academic Editor

PLOS ONE
---

## [Editor Report · Acceptance letter]

17 Jan 2020

PONE-D-19-22786R2 

Affective state determination in a mouse model of colitis-associated colorectal cancer 

Dear Dr. Mashtoub:

I am pleased to inform you that your manuscript has been deemed suitable for publication in PLOS ONE. Congratulations! Your manuscript is now with our production department. 

With kind regards,

on behalf of

Dr. Mathilde Body-Malapel 

Academic Editor

PLOS ONE